

# THE SPATIAL CONFOUNDING ENVIRONMENT

**Mauricio Tec, Ana Trisovic, Michelle Audirac**
{mauriciogtec,trisovic,maudirac,khu}@hsph.harvard.edu
Department of Biostatistics
Harvard University

**Sophie Woodward, Naeem Khoshnevis**
{swoodward, nkhoshnevis}@fas.harvard.edu
Department of Biostatistics
Harvard University

**Francesca Dominici**
fdominic@hsph.harvard.edu
Department of Biostatistics
Harvard University

## ABSTRACT

Spatial confounding poses a significant challenge in scientific studies involving spatial data, where unobserved spatial variables can influence both treatment and outcome, possibly leading to spurious associations. To address this problem, we introduce `SpaCE`: The Spatial Confounding Environment, the first toolkit to provide realistic benchmark datasets and tools for systematically evaluating causal inference methods designed to alleviate spatial confounding. Each dataset includes training data, true counterfactuals, a spatial graph with coordinates, and smoothness and confounding scores characterizing the effect of a missing spatial confounder. It also includes realistic semi-synthetic outcomes and counterfactuals, generated using state-of-the-art machine learning ensembles, following best practices for causal inference benchmarks. The datasets cover real treatment and covariates from diverse domains, including climate, health and social sciences. `SpaCE` facilitates an automated end-to-end pipeline, simplifying data loading, experimental setup, and evaluating machine learning and causal inference models. The `SpaCE` project provides several dozens of datasets of diverse sizes and spatial complexity. It is publicly available as a Python package, encouraging community feedback and contributions.

## 1 INTRODUCTION

Spatial data is paramount in several scientific domains, such as public health, social science, economics, climate science, and epidemiology. Spatial patterns may take various forms, such as geographical coordinates or network-defined adjacencies. These spatial patterns add complexity to statistical inference problems and constrain our ability to answer important questions surrounding the *causal effects* that a treatment variable has on an outcome of interest. A fundamental concern is *spatial confounding*, which occurs when an unobserved spatial variable influences the outcome and treatment simultaneously, which may lead to spurious associations between them, potentially resulting in biased estimates of causal effects (Gilbert et al., 2021b).

Consider, for example, a nationwide study estimating the effect of air pollution on cardiovascular hospitalizations. Urbanization is a confounder since it is correlated with air pollution exposure and is associated with other risks leading to higher hospitalization rates (Wu et al., 2020). If the study does not control for urban development, it may lead to incorrect estimates of the health effects of air pollution exposure. Since urbanization also varies smoothly in space (an urban area is likely to be located next to an urban area), it is a spatial confounder. In this specific scenario, one could account for urbanization using population density. However, confounders may be unknown, or measurements of a known confounder might not exist. The field of spatial confounding in causal inference aims to control for potential unobserved confounders by exploiting the spatial structure of the data, under the hypothesis that the confounding mechanism varies smoothly in space.

Spatial confounding methods seek to leverage spatial patterns to account for unobserved variables that vary smoothly in space (Gilbert et al., 2021a; Papadogeorgou et al., 2019; Tec et al., 2023). While there is an increasing interest in developing flexible machine-learning methods specifically tailored to alleviate spatial confounding (Gilbert et al., 2021b; Veitch et al., 2019). realistic benchmark datasets to measure a method's ability to reduce confounding are currently lacking. A likely reason for this absence is that counterfactuals and missing confounders cannot, by definition, be observed in real data, a challenge often known as the "fundamental problem of causal inference" (Holland, 1986).

To address this challenge, we present `SpaCE`: *The **Spa**tial **C**onfounding **E**nvironment*, a comprehensive set of software tools and datasets for evaluating machine-learning methods to address spatial confounding in causal inference. `SpaCE` datasets comprise real treatment and confounder data from publicly available sources commonly used in environmental health, social science, economics, and climate science studies, among other domains. Core to our approach is generating realistic semi-synthetic outcomes approximating true outcomes of interest using state-of-art machine-learning ensembles (Erick-

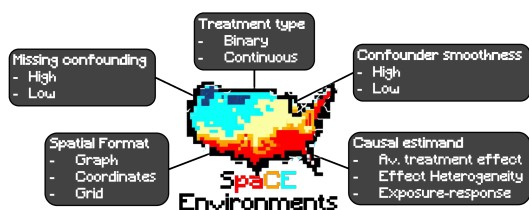

Figure 1: `SpaCE` encapsulates essential components necessary for causal effect estimation algorithms using spatial data, including ground-truth counterfactuals.

son et al., 2020) paired with cross-validation procedures for spatial data (Roberts et al., 2017) and spatial correlation modeling. In each dataset, there is a set of masked spatially-varying covariates inducing spatial confounding, and categorized by degree of spatial smoothness. `SpaCE` datasets contain spatial information in an adjacency graph and/or geographical coordinates that methods can utilize. The diversity of `SpaCE` datasets, illustrated by Fig. 1, spans multiple treatment types (binary, continuous), spatial formats (graph, coordinates), and evaluation metrics (average causal effects, dose-response curves, or counterfactual prediction at the unit level). In addition to our `SpaCE` Python package, we provide tools to reproduce existing datasets and generate new ones from a user's data.[1]

**Insufficiency of current solutions** Benchmark datasets that are currently used in causal inference lack a spatial structure and thus cannot be used for evaluating spatial confounding methods. Nonetheless, there has been an increasing interest in establishing best practices for the evaluation of causal inference methods outside of the spatial confounding literature (Cheng et al., 2022; Curth et al., 2021). There are three predominant approaches. The first one uses fully simulated treatment, confounders and outcome data by sampling from known probability distributions to verify the theoretical properties of a method in controlled settings (Morris et al., 2019). While simulation studies play an important role in developing new methods, they are insufficient for evaluating a method's performance in practice in a wide range of settings, particularly in the context of machine learning (Breiman, 2001). A second approach is to use only real data and create benchmark datasets using biased sampling from randomized controlled trials (RCTs) (Gentzel et al., 2021). However, RCTs are not available for spatial data. Additionally, RCT methods do not support the generation of counterfactuals. The third approach combines synthetic outcomes with real covariate and treatment data. These methods can be categorized as those using data-driven methods for generating a synthetic outcome (Neal et al., 2020) and those using random user-specified probability distributions without referencing actual outcome data. The latter category has the advantage of generating a large number of diverse benchmark datasets (e.g., Dorie et al. (2019)), but has recently received criticism since it lacks representativity of real data and can artificially favor arbitrary algorithms (Curth et al., 2021). The Infant Health Development Program (IHDP) dataset is the best-known example (Hill, 2011). By contrast, `SpaCE` belongs to the former category by using ensemble methods to approximate real outcomes.

## 2 BACKGROUND ON SPATIAL CONFOUNDING

We introduce some notation using the potential outcomes commonly used in the causal inference literature (Rubin, 2005). Let $A_s$ and $Y_s$ be the treatment and outcome of interest at each location $s \in \mathbb{S}$. $X_s$ indicates the vector of confounders. Boldface notation indexes spatial processes, for example, $\boldsymbol{X} = (X_s)_{s \in \mathbb{S}}$. The spatial structure in $\mathbb{S}$ is determined by a discrete graph or by a continuous

---

[1]The `SpaCE` source code is available at `https://anonymous.4open.science/r/space-BC93`

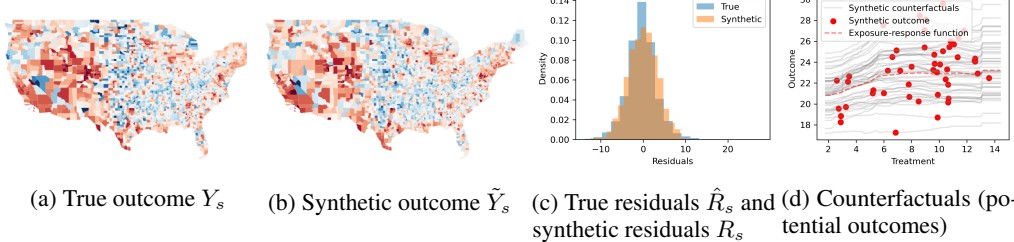

(a) True outcome $Y_s$    (b) Synthetic outcome $\tilde{Y}_s$    (c) True residuals $\hat{R}_s$ and synthetic residuals $R_s$    (d) Counterfactuals (potential outcomes)

Figure 3: Example synthetic outcome and residuals from the synthetic data generation in the `healthd_pollutn_mortality_cont` environment.

coordinate system. $Y_s^a$ represent the *potential outcomes* under a potential treatment assignment $a$ at location $s$. We also refer to potential outcomes as *counterfactuals*.

The goal of causal inference is to estimate some function of the counterfactuals that we commonly refer to as causal effects. Examples include the average treatment effect $\tau_{\text{ate}} = |\mathbb{S}|^{-1} \sum_{s \in \mathbb{S}} (\tilde{Y}_s^1 - \tilde{Y}_s^0)$ for a binary treatment, the exposure-response function $\tau_{\text{erf}}(a) = |\mathbb{S}|^{-1} \sum_{s \in \mathbb{S}} \tilde{Y}_s^a$, and the counterfactuals themselves. It is well known that when *all* confounders are observed—that is, when there are no unobserved variables affecting $A_s$ and $Y_s^a$ simultaneously—then $E[Y_s^a|X_s] = \mathbb{E}[Y_s|A_s = a, X_s]$. The left-hand side in this last expression involves counterfactuals, whereas the right-hand side can be estimated via regression. This result is known as *identification*. When a confounder is missing, identification is not guaranteed, that is, $E[Y_s^a|X_s] \neq \mathbb{E}[Y_s|X_s, A_s = a]$ (Rubin, 2005; Pearl, 2009).

*Spatial confounding* results from the co-occurrence of two conditions: there is an unobserved confounder (say, $X_s = (X_s^{\text{obs}}, X_s^{\text{miss}})$), and the process $\boldsymbol{X}^{\text{miss}}$ shows strong spatial auto-correlation. The closer locations $s$ are $s'$ are, the more correlated $X_s^{\text{miss}}$ and $X_{s'}^{\text{miss}}$ become (as illustrated in Fig. 2). A pronounced spatial distribution of a missing confounder suggests a similar confounding nature between nearby locations $s$ and $s'$. This lack of identification is what methods for spatial confounding aim to address, exploiting knowledge of the spatial structure. While this goal may be achieved in different ways in different methods, the general strategy can intuitively be described as learning some function of space $Z_s$ so that identification holds, that is, $E[Y_s^a|X_s, Z_s] \approx E[Y_s|A_s = a, X_s, Z_s]$.

As a technical remark, the spatial confounding literature typically assumes that the treatments of one unit do not affect the outcomes of other units, a condition known as *interference* (Forastiere et al., 2021). Much like spatial confounding, interference can bias causal estimates. However, their underlying mechanisms and solutions are distinct (Ogburn & VanderWeele, 2014; Gilbert et al., 2021b; Papadogeorgou & Samanta, 2023). It is worth noting that failing to address spatial confounding may give the false impression of interference even if the latter is not present in the data (Papadogeorgou & Samanta, 2023). Here, we focus on spatial confounding, deferring interference to future work (we provide additional discussion in Section 6).

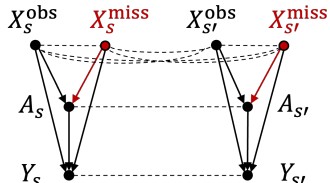

Figure 2: Causal diagram of spatial confounding with neighbors $s$ and $s'$. Arrows represent causal relations; undirected dotted lines represent non-necessarily causal associations. The correlations increase as the distance between $s$ and $s'$ decreases.

Lastly, notice that the term spatial confounding has been used inconsistently across research, sometimes without a clear causal inference interpretation (Gilbert et al., 2021b; Khan & Berrett, 2023). For instance, spatial confounding has sometimes been characterized as a property of statistical *models* (Dupont et al., 2022), specifically, as the correlation between the spatial random effects of a model and the treatment variable. Another definition has been concerned primarily with *variance* adjustments for spatially autocorrelated data (Khan & Calder, 2022; Reich et al., 2006).

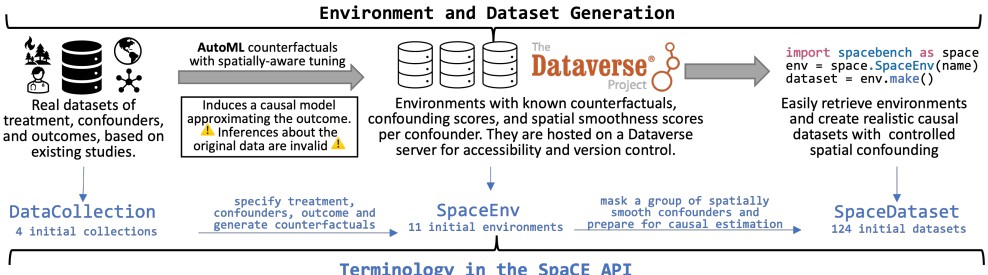

Figure 4: *(Top):* The semi-synthetic data generation pipeline and acquisition. *(Bottom):* Summary of the key terms used in `SpaCE`.

# 3 `SpaCE`: THE SPATIAL CONFOUNDING ENVIRONMENT

We first explain the design goals and overview of the `SpaCE` pipeline and its components. Next, we provide details about the various implementations of these components.

**Design goals**  The first requirement is that the generated datasets are representative of real data. We will achieve this by using real covariate and treatment data with realistic, semi-synthetic outcomes and counterfactuals. We will provide details later in this section. Data generating mechanisms producing datasets that could artificially favor a specific model type during causal evaluation should be ideally avoided (Curth et al., 2021). The spatial autocorrelation of the residuals (the unexplained variation in the outcome) should also be represented accurately since various spatial confounding adjustment methods are based on adding "spatial effects" to a predictive model (Dupont et al., 2022), and their performance in resolving confounding is affected by the spatial autocorrelation pattern in the data.

**Overview of the pipeline**  The `SpaCE` pipeline consists of two stages: (1) generate a synthetic data *environment* by selecting a treatment, outcome, and covariates from a publicly available data *collection* based on a realistic scientific question; and (2) obtain a benchmark *dataset* with controlled spatial confounding by masking a group of spatial covariates in the environment. To illustrate the idea with an example, let us consider the treatment is air pollution exposure and the outcome is hospitalizations. First, we will fit a predictive model using air pollution and a set of observed confounders, such as socio-demographic characteristics, including urbanization. For the residuals, we will then use a probabilistic model to obtain an independent sample of a spatial process approximating the distribution of this model's errors. Finally, we would create a benchmark dataset by masking a group of confounders, such as urbanization variables. This example is present in `SpaCE`, with the generated data displayed in Fig. 3, showing the counterfactuals, outcomes, and residuals alongside the real data, highlighting their striking resemblance.

**Key terminology**  We introduce three key terms that will facilitate the presentation of our data generation pipeline and are used throughout the `SpaCE` API. These terms are illustrated in Fig. 4. First, a `DataCollection` comprises variables from publicly available data surrounding a theme or topic of interest. From each of these collections, we can obtain various causal inference environments, or `SpaceEnvs`, after specifying a treatment, outcome, and set of confounders—emulating a realistic study—and implementing our data fitting and generation method. Each environment contains semi-synthetic counterfactuals with important metadata, such as a list of edges or geographic coordinates. Environments are stored on a server where it is publicly accessible, versioned, and documented. Lastly, each environment yields various benchmarking datasets, or `SpaceDatasets`, obtained by masking a group of related confounders varying smoothly in space. Every `SpaceDataset` contains metadata about the degree of spatial smoothness and a "confounding score" determined by their respective omitted group of confounders.

**Data Collections**  `DataCollections` are the entry-point to the semi-synthetic data-generating algorithm. Currently, `SpaCE` offers six `DataCollections` from various domains and sizes, summarized below and in Table 1. Appendix A provides additional details about the public data sources composing each collection.

The smallest dataset represents a spatial graph of 3,109 nodes, while the largest has nearly 4 million nodes. Notice, however, that the size of the smaller datasets may already be computationally

Table 1: List of `DataCollection`s and `SpaceEnv`s A `SpaceEnv` name has four components: the prefix is the code for the source `DataCollection` (`cdcsvi`, `climate`, `healthd`, `county`), it is followed by the treatment and outcome codes, and ends with the code for the treatment type (`cont`, `disc`).

| DataCollection / SpaceEnv | Treatment Type | Spatial Unit | Region | #Covariates #SpaceDatasets | | #Nodes | #Edges |
|---|---|---|---|---|---|---|---|
| Social Vulnerability and Welfare | | census tract | Texas | 16 | | 6,828 | 21,585 |
| `cdcsvi_limteng_hburdic_cont` | continuous | | | | 12 | | |
| `cdcsvi_nohsdp_poverty_cont` | continuous | | | | 11 | | |
| `cdcsvi_nohsdp_poverty_disc` | binary | | | | 11 | | |
| Heat Exposure and Wildfires | | census tract | California | 22 | | 8,616 | 26,695 |
| `climate_relhum_wfsmoke_cont` | continuous | | | | 8 | | |
| `climate_wfsmoke_minrty_disc` | binary | | | | 10 | | |
| Air Pollution and Mortality | | county | USA | 34 | | 3,109 | 9,237 |
| `healthd_dmgrcs_mortality_disc` | binary | | | | 9 | | |
| `healthd_hhinco_mortality_cont` | continuous | | | | 10 | | |
| `healthd_pollutn_mortality_cont` | continuous | | | | 9 | | |
| Welfare and Elections | | county | USA | 45 | | 3,109 | 9,237 |
| `county_educatn_election_cont` | continuous | | | | 14 | | |
| `county_phyactiv_lifexpcy_cont` | continuous | | | | 16 | | |
| `county_dmgrcs_election_disc` | binary | | | | 14 | | |
| PM$_{2.5}$ Components | | $1 \times 1$ km | USA | 6 | | 3,882,956 | 7,647,552 |
| `pm25_hires_no3_pm_cont` | continuous | | | | 4 | | |
| `pm25_hires_so2_pm_cont` | continuous | | | | 4 | | |
| Socioeconomic Status and Broadband Usage | | zip code | USA | 16 | | 30,383 | 174,345 |
| `zcta_income_broadband_disc` | discrete | | | | 5 | | |
| `zcta_age_broadband_cont` | continuous | | | | 4 | | |

demanding for some spatial confounding algorithms with polynomial space complexity. In our experiments, we found this to be true using many practical implementations of spatial regression methods, such as those based on the pysal library (Rey & Anselin, 2007). Further, the literature on spatial confounding has focused strongly on developing $\sqrt{n}$-consistent estimators that have quick convergence with high performance on similarly sized or smaller datasets (Gilbert et al., 2023). Notably, our benchmark dataset sample sizes surpass those in many simulation studies of these methods. On the other hand, the largest datasets can be interesting for the performance of scalable methods, such as neural networks.

## 4 DETAILS ABOUT THE DATA GENERATION PIPELINE

We now provide technical details about the data-generating mechanism in `SpaCE`.

**Spatial Semi-synthetic Data Generation: From `DataCollection` to `SpaceEnv`** Given a `DataCollection`, we select a set of treatment, outcome, and covariates to represent a realistic scientific question. Each combination produces a semi-synthetic outcome and counterfactuals, resulting in a unique `SpaceEnv`. Let $\boldsymbol{X}$, $\boldsymbol{A}$, and $\boldsymbol{Y}$ denote the confounders, treatment, and outcome. We approximate the distribution of the outcome as $\tilde{Y}_s = f(X_s, A_s) + R_s$ such that $\Pr(\boldsymbol{Y}|\boldsymbol{A}, \boldsymbol{X}) \approx \Pr(\tilde{\boldsymbol{Y}}|\boldsymbol{A}, \boldsymbol{X})$. We learn $f$ and $\boldsymbol{R}$ for each `SpaceEnv`. We can then generate counterfactuals by evaluating on any treatment value:

$$\tilde{Y}_s^a = f(X_s, a) + R_s \tag{1}$$

The strategy to generate counterfactuals has four steps. First, we learn $f$ that best predicts $Y_s$ from $(X_s, A_s)$ using AutoML. Second, we estimate the empirical additive errors $\hat{R}_s = Y_s - f(X_s, A_s)$ and their joint (spatial) distribution $\hat{\boldsymbol{R}} \sim P_R$. Third, we replace these endogenous residuals with an independent similarly distributed exogenous noise $\boldsymbol{R} \sim P_R$. Finally, we obtain counterfactuals with Eq. (1) by varying the treatment while holding constant the confounders and the exogenous noise. The methodology implementing these steps pursues the following desirable properties: (a) $f$ captures non-linear relations and interactions; (b) $\boldsymbol{R}$ exhibits a similar spatial autocorrelation as the one observed in the data when $f$ captures all the causal relations from $(\boldsymbol{X}, \boldsymbol{A})$ to $\boldsymbol{Y}$; (c) the resulting dataset does not exhibit additional unobserved confounding. Sampling $\boldsymbol{R}$ independently ensures that the synthetic residuals are independent of the treatment, and therefore, the only possible confounders are those in $\boldsymbol{X}$ by the backdoor criterion (Pearl, 2009). Below, we provide more details about the implementation.

*Predictive model.* For (a), we learn $f$ using ensembles of machine-learning models where the ensemble weights are determined by their predictive ability on validation data. We do this to

avoid favoring causal algorithms based on a specific type of model, as recommended by Curth et al. (2021), who discuss best practices for causal inference benchmarks. We fit the ensemble using the `AutoGluon` Python package (Erickson et al., 2020) to reduce human intervention, automatically selecting the best hyperparameters and performing various overfitting reduction techniques. The package options and configuration are described in Table 5 in the supplement. We found it *critical* to implement a *spatially-aware* train-validation data split (Roberts et al., 2017), resulting in extreme overfitting without it, caused by the duplicity of training data in the validation set due to spatial correlations. The spatially-aware splitting scheme selects a small subset of validation nodes and uses breadth-first search to remove their surrounding neighbors from the training subset. This algorithm is described in Algorithm 1 in the supplement.

*Residual sampling.* For (b) and (c), residual sampling, we use a Gaussian Markov Random Field (GMRF) from a spatial graph, a convenient and scalable spatial process (Rue & Held, 2005; Tec et al., 2019). More precisely, we sample the synthetic residuals as $\boldsymbol{R} \sim_{\text{iid}} \text{MultivariateNormal}(\boldsymbol{0}, \hat{\lambda}(\mathbb{D} - \hat{\rho}\mathbb{A}\mathbb{D})^{-1})$, where $\mathbb{A}$ is the adjacency matrix of the spatial graph; $\mathbb{D}$ is a diagonal matrix with the number of neighbors of each location; $\hat{\rho}$ determines the correlation between an observation and its neighbors, learned from the true residuals (the errors of the predictive model in Step 1); and $\hat{\lambda}$ is chosen to match the variance of the true residuals exactly. A key technical element in our implementation was sampling from a large Gaussian random field using matrix factorizations for sparse matrices (Chen et al., 2008), enabling scalability to massive spatial graphs (such as in the *PM$_{2.5}$ Components* `DataCollection`).

Panel (c) in Fig. 3 illustrated the marginal distribution of the fitted residuals compared with the original ones. Further, panel (d) showed the fitted counterfactuals, which are visualized as functions of the treatment/exposure passing through the observed data points. Fig. 6 in the appendix provides additional visualizations comparing the synthetic and real residuals. In particular, panel (b) compares the Moran I measure for spatial autocorrelation across all of the `SpaCE` datasets, indicating a strong correspondence validating the learning procedure.

Table 2: Contents of a `SpaceDataset`.

| Name | Description |
|---|---|
| Training data | $\boldsymbol{X}^{\text{obs}}$, $\boldsymbol{A}$ and $\tilde{\boldsymbol{Y}}$ |
| Counterfactuals | $\tilde{\boldsymbol{Y}}^a$ for each treatment value $a \in \mathbb{A}$, computed with Eq. (1). For continuous treatments, a discretization of $|\mathbb{A}| = 100$ values is used |
| Graph and coordinates | A list of edges and coordinate matrix |
| Treatment type | An indicator if $|\mathbb{A}| = 2$ (binary) or $|\mathbb{A}| = 100$ (continuous) |
| Spatial smoothness score | Auto-correlation of $\boldsymbol{X}^{\text{miss}}$ |
| Confounding score | A metric of the degree of confounding induced by $\boldsymbol{X}^{\text{miss}}$. . |

**Inducing Spatial Confounding: From** `SpaceEnv` **to** `SpaceDataset` A benchmark dataset is obtained by masking a group of related confounders in a `SpaceEnv`. Masking entails removing a group of related covariates from the training data. A dataset object encapsulates the essential components required for estimating causal effects and comparing the performance of spatial confounding methods. The contents of a `SpaceDataset` are summarized in Table 2. The variable names in the space datasets are *anonymized* to discourage inappropriate use of our benchmark datasets to infer effects in the original data collections. We provide additional discussion in Section 6.

Eq. (1) specifies an additive noise model (ANM) used for counterfactual generation. While this model is additive, `SpaceDatasets` can exhibit more complex forms of non-additive noise due to the interactions with the missing confounder. Nonetheless, it is worth noticing that ANMs remain the focus and leading open challenge in the spatial confounding literature (c.f., Akbari et al. (2023)). Even when outcomes are count-based or binary, which could indicate non-additive noise, simple techniques from generalized linear models (e.g., Poisson regression) or outcome transformations (e.g., logarithms) have been adapted to complement spatial confounding methods (c.f., Urdangarin et al. (2023)). Thus, `SpaceDatasets` are highly relevant to the current state of the art in spatial confounding adjustment. The ANM will be a sensible model when the empirical residuals are approximately symmetric and continuously supported, as exemplified in Fig. 3c. Documentation of the outcome transformations and histograms of the empirical and synthetic residuals are stored with each `SpaceEnv` in the Harvard Dataverse platform.

**Tasks and Metrics** Tasks in `SpaCE` are determined by a causal effect target which, in turn, determines the metrics used to evaluate performance. Not all algorithms can estimate all types of causal effects, and the relevant effects also depend on whether the treatment is binary or continuous. Section 4 summarizes the causal effect targets currently supported in `SpaCE` and the associated metrics which can be computed from the counterfactuals. `SpaCE` provides an auxiliary class called

Table 3: Supported causal estimation tasks and metrics. $\tau_*$ denotes a causal effect and $\hat{\tau}_*$ is an estimate, where $*$ is the name of the task. For the last column, $\hat{Y}_s^a$ is an estimate of the counterfactual $\tilde{Y}_s^a$ for each location $s$ and treatment $a$. For continuous treatment types, we assume $|\mathbb{A}| = 100$. The metrics are standardized by the standard deviation of $\tilde{Y}_s$, denoted $\sigma_y$.

| Causal Effect | Treatment type | Metric |
|---|---|---|
| average treatment effect (ATE) $\tau_{\text{ate}} = \|\mathbb{S}\|^{-1} \sum_{s \in \mathbb{S}} (\tilde{Y}_s^1 - \tilde{Y}_s^0)$ | binary | absolute bias (BIAS) $\sigma_y^{-1} \|\hat{\tau}_{\text{ate}} - \tau_{\text{ate}}\|$ |
| exposure-response function (ERF) $\forall a \in \mathbb{A} : \tau_{\text{erf}}(a) = \|\mathbb{S}\|^{-1} \sum_{s \in \mathbb{S}} \tilde{Y}_s^a$ | continuous | root mean integrated-squared error ($\sqrt{\text{MISE}}$) $\sigma_y^{-1} \sqrt{\|\mathbb{A}\|^{-1} \sum_{a \in \mathbb{A}} (\hat{\tau}_{\text{erf}}(a) - \tau_{\text{erf}}(a))^2}$ |
| individualized treatment effects (ITE) $\forall a \in \mathbb{A}, s \in \mathbb{S} : \hat{Y}_s^a$ | binary and continuous | precision at estimating heterogeneous effects (PEHE) $\sigma_y^{-1} \sqrt{\|\mathbb{A}\|^{-1} \|\mathbb{S}\|^{-1} \sum_{s \in \mathbb{S}, a \in \mathbb{A}} (\tilde{Y}_s^a - \hat{Y}_s^a)^2}$ |

`DatasetEvaluator` to estimate these quantities. The performance metrics are standardized by the standard deviation of the synthetic outcome so that they are comparable across datasets and environments.

The three causal estimands currently supported in `SpaCE` are the average treatment effect (ATE) (for binary treatments), the exposure-response function (ERF) (also known as the average dose-response curve), and the individualized treatment effects (ITE). The value of these metrics is defined in terms of potential outcomes in the first column of Section 4. The ATE and ERF are averaged across the population, while the ITE is simply the counterfactual value for each unit. We will measure performance based on metrics commonly used in the machine learning literature for these estimands (c.f., (Cheng et al., 2022)): the absolute bias BIAS for the ATE (Hill, 2011; Shi et al., 2019); the mean integrated squared error MISE for the ERF (Schwab et al., 2020; Nie et al., 2020); and the precision in estimating heterogeneous effects PEHE for the ITE (Hill, 2011; Shi et al., 2019). The names of these metrics vary in the literature, but the definitions are consistent or present small variations. The standard deviation of the outcome normalizes all effects so they are comparable across datasets.

**Confounding score and smoothness scores** These scores characterize the properties of the masked confounders of a `SpaceDataset`. Smoothness scores are calculated using Moran's I statistic, which is an approximate form of spatial correlation and takes values in $[-1, 1]$ (Moran, 1950). The confounding score is computed as a baseline model's change in causal estimates when masking a set of confounders. We use the same AutoML configuration of the predictive model as the baseline. Each causal estimand defined in Section 4 has an associated confounding score for each benchmark dataset. For ease of presentation, we use the confounding score of the exposure-response function as the single reference. We will also classify datasets in low or high confounding/smoothness, using the median across all datasets as the threshold.

## 5 EXAMPLES AND EXPERIMENTS

Our first experiment investigates whether or not the generated `SpaceDatasets` have the property that (1) the causal effects can be learned from the generated semi-synthetic outcomes and (2) masking confounders reduces the estimation performance. To assess this, we collected the estimation errors of three baselines when using the unmasked full covariates and when learning with masked spatial confounders based on the `cdcsvi_nohsdp_poverty_cont` environment. The baselines considered are ordinary least-squares (OLS), gradient boosting (XGBOOST), and a multi-layer perception (MLP). We did not tune for the best architecture in this experiment since the purpose was not to discover the best method but to evaluate points (1) and (2) just mentioned.

The results of the experiment are shown in Fig. 5. In all cases, training with the masked `SpaceDatasets` decreases performance compared to using all the confounders. The MLP is the best performer, also corroborated by the examples of fitted curves in panel (a), which are reasonably acurate. Another takeaway is that a method cannot perfectly estimate the true causal effects due to model limitations and finite sample error, even when using all the confounders.

Our second experiment illustrates the evaluation of spatial confounding methods. It is worth noting that linear models have dominated this literature without strict attention to scalability. Thus, due to

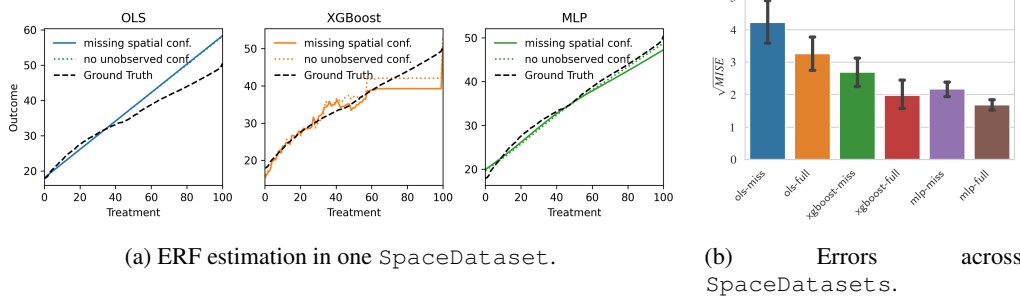

(a) ERF estimation in one `SpaceDataset`.    (b) Errors across `SpaceDatasets`.

Figure 5: Examples using the `cdcsvi_nohsdp_poverty_cont` environment.

limitations in the scalability of some of the baselines taken into account below, we only conducted this experiment on the first four data collections in Section 4.

We first include two baseline models from the PySAL Python library (Rey & Anselin, 2007). These models are the Spatial Two Stage Least Squares (Anselin, 1988) (S2SLS), implementing an outcome auto-regressive model, and the General Methods of Moments Estimation of the Spatial Error (Kelejian & Prucha, 1999; 1998) (GLMERR), implementing a residual auto-regressive model. Next, we consider the SPATIAL and SPATIAL+ models, which use spatial splines and a two-stage regression procedure to account for the spatial portion of the treatment (Dupont et al., 2022). We also consider a Graph Convolutional Neural Network (Kipf & Welling, 2016) (GCNN), which uses neighboring information similarly to S2SLS but can also model non-linear relations from a potentially longer range of spatial dependencies. Thus, while not typically considered a spatial confounding method, it is worth evaluating its performance. Next, we consider the distance-adjusted propensity score method (DAPSM) (Papadogeorgou et al., 2019), which uses a matching estimator (Rubin, 1973) that prioritizes nearby matches. This method finds pairs of treated and untreated units to compare their outcomes as an estimate of the causal effect (only applicable in the binary case).

To ensure a fair comparison, we use the RAY TUNE (Liaw et al., 2018) framework for hyperparameter tuning. In all cases but DAPSM, we use the out-of-sample prediction as the tuning criterion. For DAPSM we use the covariate balance criterion following Papadogeorgou et al. (2019). Appendix D provides additional details about the tuning hyperparameters of each method. Importantly, we use the spatially-aware train-validation splitting for computing the tuning metric since random splitting would result in extreme overfitting (Roberts et al., 2017).

Table 4 shows the results, which vary strongly across settings. Notably, the SPATIAL+ algorithm excels in estimating binary average treatment effects yet underperforms in continuous treatment scenarios. In the latter case, the GMERROR seems to yield the best performance, but without statistical significance. Another interesting result is that linear models consistently outperform GCNN, even when they cannot inherently account for effect heterogeneity as the neural network. Although the GCNN could perform well in various experiments, it would sometimes lead to large errors, affecting the final metric. In general, models highly sensitive to overfitting were more challenged by the challenges of hyper-parameter tuning due to the spatial correlations.

Fig. 10 in the appendix summarizes the experiment results grouped by `SpaceEnv`. One takeaway is that the linear spatial regression baselines GMERR and SL2SLS offer only a very small improvement over a baseline OLS in all environments. This result could indicate the need to focus on expanding the literature on machine-learning-driven models. We can see that the GCNN had a better performance than the other models in the two environments from the *Heat Exposure and Wildfires* collection (see Table 1) despite its overall worst performance, warranting future investigation of what properties of this collection allowed for a better performance of the GCNN.

Lastly, we analyzed the baselines' performance by smoothness and confounding score. For this purpose, we use a linear mixed effects model (Gałecki et al., 2013). See Appendix F for additional details of the model specification. The results of this analysis, shown in Table 9 of the appendix, indicate that the errors were lower as the smoothness score was higher for all baselines. DAPSM benefited the most from higher smoothness. Conversely, a higher confounding score is associated with higher errors in all but one case. Many of the effects are moderately statistically significant. This analysis provides additional evidence that the smoothness and confounding scores are meaningful in describing the complexity of a benchmark dataset, although with high variance.

Table 4: Benchmarks are collected across all datasets. Table entries show the mean with 95% confidence intervals using the asymptotic normal formula. "High smoothness" and "high confounding" denote datasets in the top 50% based on spatial smoothness and confounding score. Sample sizes, depicted in Fig. 8, reflect dataset combinations of smoothness, confounding, and treatment type.

| SMOOTHNESS | CONFOUNDING | METHOD | BINARY TREATMENT | | CONTINUOUS TREATMENT | |
| --- | --- | --- | --- | --- | --- | --- |
| | | | ATE | ITE | ERF | ITE |
| HIGH | HIGH | DAPSM | 0.29 ± 0.20 | 0.25 ± 0.15 | N/A | N/A |
| | | GCNN | 0.16 ± 0.07 | 0.25 ± 0.08 | 0.49 ± 0.14 | 0.72 ± 0.16 |
| | | GMERROR | 0.07 ± 0.04 | 0.14 ± 0.05 | **0.26 ± 0.07** | 0.45 ± 0.11 |
| | | S2SLS | 0.11 ± 0.06 | 0.15 ± 0.07 | 0.26 ± 0.07 | 0.45 ± 0.11 |
| | | SPATIAL+ | **0.03 ± 0.03** | **0.13 ± 0.05** | 0.42 ± 0.13 | 0.56 ± 0.14 |
| | | SPATIAL | 0.05 ± 0.03 | 0.13 ± 0.06 | 0.26 ± 0.08 | **0.44 ± 0.11** |
| | LOW | DAPSM | 0.39 ± 0.16 | 0.37 ± 0.09 | N/A | N/A |
| | | GCNN | 0.14 ± 0.04 | 0.28 ± 0.04 | 0.44 ± 0.11 | 0.58 ± 0.12 |
| | | GMERROR | 0.05 ± 0.02 | **0.18 ± 0.04** | **0.37 ± 0.19** | 0.43 ± 0.19 |
| | | S2SLS | 0.04 ± 0.02 | 0.18 ± 0.04 | 0.86 ± 1.01 | 0.94 ± 1.06 |
| | | SPATIAL+ | **0.03 ± 0.01** | 0.18 ± 0.04 | 0.41 ± 0.19 | 0.47 ± 0.19 |
| | | SPATIAL | 0.04 ± 0.01 | 0.18 ± 0.04 | **0.37 ± 0.19** | **0.42 ± 0.19** |
| LOW | HIGH | DAPSM | 0.18 ± 0.11 | 0.19 ± 0.10 | N/A | N/A |
| | | GCNN | 0.14 ± 0.06 | 0.18 ± 0.08 | 0.31 ± 0.08 | 0.57 ± 0.07 |
| | | GMERROR | 0.09 ± 0.02 | 0.16 ± 0.06 | **0.31 ± 0.07** | **0.48 ± 0.10** |
| | | S2SLS | 0.09 ± 0.01 | 0.16 ± 0.06 | 0.31 ± 0.07 | 0.48 ± 0.10 |
| | | SPATIAL+ | **0.05 ± 0.03** | **0.15 ± 0.06** | 0.47 ± 0.12 | 0.60 ± 0.13 |
| | | SPATIAL | 0.06 ± 0.03 | 0.15 ± 0.06 | 0.31 ± 0.07 | 0.48 ± 0.10 |
| | LOW | DAPSM | 0.53 ± 0.15 | 0.45 ± 0.08 | N/A | N/A |
| | | GCNN | 0.17 ± 0.04 | 0.34 ± 0.03 | 0.36 ± 0.09 | 0.50 ± 0.09 |
| | | GMERROR | 0.06 ± 0.02 | **0.18 ± 0.04** | **0.32 ± 0.19** | **0.37 ± 0.19** |
| | | S2SLS | 0.08 ± 0.05 | 0.19 ± 0.05 | 0.70 ± 0.57 | 0.77 ± 0.59 |
| | | SPATIAL | 0.05 ± 0.02 | 0.18 ± 0.04 | 0.34 ± 0.21 | 0.39 ± 0.20 |
| | | SPATIAL+ | **0.03 ± 0.01** | 0.18 ± 0.04 | 0.35 ± 0.19 | 0.40 ± 0.19 |

# 6    CONCLUSION AND DISCUSSION

By enabling a systematic evaluation and comparison of spatial confounding in realistic scenarios, SpaCE represents a significant step towards addressing spatial confounding, a fundamental issue in many scientific problems with spatial data. SpaCE advances causal inference in spatial data by providing a well-structured, adaptable environment for benchmarking and improving existing and future methods. We will continuously grow the available environments in the SpaCE Python package to include additional DataCollections from various domains.

There are exciting opportunities for future work. One possible extension is temporal confounding, which shares mechanisms with spatial confounding when a confounder has lagged effects and varies smoothly in time (Imai et al., 2021; Papadogeorgou et al., 2022). Our methods to generate semi-synthetic data may be reused. In particular, our approach to function estimation could use time series methods and our residual generation could be generalized to a spatio-temporal graph (Tec et al., 2019). However, the methods to resolve temporal and spatial confounding can differ significantly. For instance, difference-in-differences methods are only used with longitudinal data. Similarly, baselines will be different, as well as residual sampling strategies.

Another possible extension is *interference*, also known as *spill-overs*, in which the treatment of neighbors affects the outcomes of other units. However, significant additional work is needed for this extension. First, ensembles of mainstream machine learning models (e.g. boosting, random forests, MLPs) are not suitable for data with unknown interference (Bhattacharya et al., 2020). GNNs are promising tools, but much evaluation is still needed (Ma & Tresp, 2021), and, for best practices, ensembling methods should not rely on a single type of generative model (Curth et al., 2021). Second, the interference literature focuses on estimating spillover effects (Hudgens & Halloran, 2008). Metrics and evaluation tools for this task need careful design. Third, the data domains of interest in the spill-overs literature may vary. For instance, it has concentrated more often on social networks (Forastiere et al., 2021), while they have been of less interest in spatial confounding.

Addressing these future work opportunities here would have required substantial developments that are beyond the scope of spatial confounding. It must be highlighted that some solutions for these extensions already exist (Cheng et al., 2022), while SpaCE addresses a fundamental gap in spatial confounding, for which no solution with realistic data currently exists.

ETHICS STATEMENT AND BROADER IMPACT

We consider several ethical aspects while designing and deploying `SpaCE`. First, we take several actions to minimize potential harm due to misusing semi-synthetic datasets to derive conclusions and inferences about causal effects in real data: we (1) anonymize variable names in `SpaceDataset`, (2) issue warning statements clarifying limitations when users import the Python package, and (3) emphasize the synthetic nature the data in the description of each environment in a data repository where they are available publicly. Second, to increase transparency and reproducibility, we provide detailed instructions for full reproducibility, including publicly available and accessible code. Third, while our work does not directly address concerns about fairness and societal biases, our data collections integrate sources commonly used in studies of social vulnerability and health equity, driving methodology improvement for social good.

ACKNOWLEDGMENTS

This work was supported by the NIH awards RF1AG080948-S1 (MT, NK), R01ES029950 (MT), 1R01MD016054-01A1 (FD), 5R01ES030616-04 (MT), 3RF1AG074372-01A1S1 (MT), P30ES000002 (MT), and R01MD016054-S2 (MT); NIEHS training grants T32ES007142 (SW) and T32ES007069 (KH); Alfred P. Sloan Foundation grant G-2020-13946 (FD, KH); and the Fernholz Foundation (MT). The computations in this paper were performed on the FASRC Cannon clusters supported by the FAS Division of Science Research Computing Group at Harvard University.

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

APPENDIX

## A    DATA COLLECTIONS

We provide additional details of each `DataCollection`. See also the *Code and API* paragraph in the next section describing their documentation in the package repository.

The *Social Vulnerability and Welfare* collection covers data related to the Social Vulnerability Index (SVI) provided by the Agency for Toxic Substances and Disease Registry (ATSDR) (Centers for Disease Control and Prevention, 2020). The SVI is a widely used index provided by the Centers for Disease Control (CDC). This collection is built with census tract-level variables for the state of Texas, USA, and incorporates data on unemployment, racial and ethnic minority status, and disability (US Census Bureau, 2010).

The *Heat Exposure and Wildfires* collection includes information about the monthly weather and climate conditions during the summer of 2020 in California. It covers factors like temperature, wildfire smoke, wind speed, and relative humidity and population density aggregated in the U.S. Census tract level. These are sourced from Parameter-elevation Regressions on Independent Slopes Model (PRISM) (PRISM Climate Group), the ambient wildfire smoke $PM_{2.5}$ model (Childs et al., 2022), and National Oceanic and Atmospheric Administration (NOAA).

The *Air Pollution and Mortality* collection incorporates mortality data from respiratory and cardiovascular diseases among the elderly sourced from the CDC, along with air pollution exposure variables (Di et al., 2019), risk factors from BRFSS (CDC, 2010), and Census data (US Census Bureau, 2010) covering the mainland US at the county level for 2010.

The *Welfare and Elections* collection is a subset of the US County data on election outcomes, specifically the percentage of the population in each county that voted for the Democratic Party in 2020 (McGovern & Morris, 2016). It is expanded with demographics, such as educational attainment and health (University of Wisconsin Population Health Institute, 2021; Glasmeier, Amy K. Living Wage Calculator, 2020; Institute for Health Metrics and Evaluation, 2014), crime (Washington Post, 2020), and employment statistics (Bureau of Labor Statistics, 2019), from 2019 and 2020.

The *PM$_{2.5}$ Components* collection combines the high resolution total $PM_{2.5}$ dataset of Di et al. (2019) with the $PM_{2.5}$ composition data from Amini et al. (2022), both available at high spatial resolution of $1 \times 1$ km grid. We use the datasets corresponding to the annual average for 2000.

The *Socioeconomic Status and Broadband Usage* collection obtains 2010 zip-code level broadband usage from Pereira et al. (2021) and combines it with socioeconomic status variables from the census (US Census Bureau, 2010).

## B    `SpaceEnv` GENERATION DETAILS

This section explains the training procedure for obtaining the `SpaceEnv` from a `DataCollection`.

Recall from Section 3 that we learn a generative model of outcome and obtain semi-synthetic counterfactuals as $\tilde{Y}_s^a = f(X_s, a) + R_s$ where $f$ is obtained using AutoML and $\boldsymbol{R} = (R_s)_{s \in \mathbb{S}}$ is learned as a Gaussian Markov random field matching the spatial distribution of the observed residuals in the training data. Below, we describe each step of the environment generation proceedure and the API that a user can use to generate new `SpaceEnvs`.

**Training $f$ using AutoML**    The predictor $f$ is obtained using auto machine-learning (AutoML) techniques implemented with the AutoGluon package in Python (Erickson et al., 2020). This package trains an ensemble of models and computes a weighted ensemble where the weights are based on the cross-validation performance. Table 5 describes the default settings used for Autogluon. It is important to note that our models have minimal manual tuning, since we aim that training with real data drives the generated synthetic outcomes.

We used a special train-validation split since the default random split led to extreme overfitting caused by spatial correlations. Intuitively, spatial correlations create duplicates in the data. As a

Table 5: Hyperparameters used in AutoML

| Parameter | Value |
|---|---|
| `package` | `AutoGluon v0.7.0` |
| `fit.presets` | `good_quality` |
| `fit.tuning_data` | custom with Algorithm 1 |
| `fit.use_bag_holdout` | false |
| `fit.time_limit` | 3600 |
| `feature_importance.time_limit` | 3600 |

result, a random split creates almost identical copies in the train and validation sets. This means the cross-validation procedures will always choose algorithms that overfit more since they would minimize the error on the identical copies in the validations set.

To solve over-fitting due to spatial correlations, our spatially-aware validation splitting algorithm is explained in Algorithm 1, using breadth-first search (BFS) to obtain a spatially contiguous validation set with a buffer. The algorithm relies on specifying a number of initial seeds for the validation set obtained with random sampling. It expands the validation set with a specified number of BFS neighbors. It removes additional BFS levels from training and validation. Using the default parameters specified in Algorithm 1, we consistently obtain training splits of size $50\% - 70\%$ and validations splits of size $10\% - 20\%$. We recommend checking if adjusting these values is required when using new data collections.

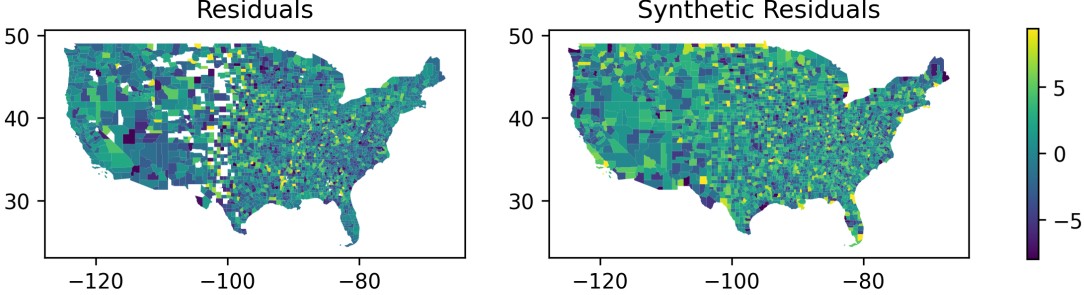

(a) A comparison of the true and synthetic residuals on the `healthd_hhinco_mortality_cont` environment. Coordinates represent latitude and longitude. Missing data is shown in white in the left figure. The synthetic residuals can be sampled in missing are regions because the treatment and confounders are available.

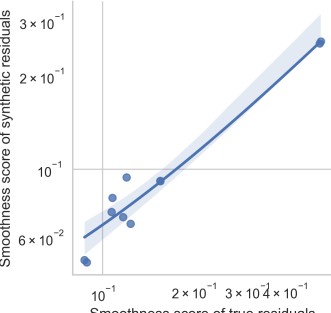

(b) A comparison of the spatial smoothness (measured by Moran's I) between the true residuals from the AutoML predictive model and the synthetic residuals from the Gaussian Markov Random Field.

Figure 6: Additional visualizations of the synthetic residuals.

**Sampling $R$ using a Gaussian Markov Random Field** We sample the synthetic residuals as

$$R \sim_{\text{iid}} \text{MultivariateNormal}(\mathbf{0}, \hat{\lambda}(\mathbb{D} - \hat{\rho}\mathbb{A}\mathbb{D})^{-1}),$$

---

**Algorithm 1** Spatially-aware validation split selection

---

**Input:** Graph as map of neighbors $s \to \mathbb{N}_s$ where $\mathbb{N}_s \subset \mathbb{S}$ is the set of neighbors of $s$.
**Params:** Fraction $\alpha$ of seed validation points (default $\alpha = 0.02$); number of BFS levels $L$ to include in the validation set (default $L = 1$); buffer size $B$ indicating the number of BFS levels to leave outside training and validation (default $B = 1$).
**Output:** Set of training nodes $\mathbb{T} \subset \mathbb{S}$ and validation nodes $\mathbb{V} \subset \mathbb{S}$.
 1: *# Initialize validation set with seed nodes*
 2: $\mathbb{V} = \text{SampleWithoutReplacement}(\mathbb{S}, \alpha)$
 3: *# Expand validation set with neighbors*
 4: **for** $\ell \in \{0, \dots, L - 1\}$ **do**
 5:     $\text{tmp} = \mathbb{V}$
 6:     **for** $s \in \text{tmp}$ **do**
 7:         $\mathbb{V} = \mathbb{V} \cup \mathbb{N}_s$
 8:     **end for**
 9: **end for**
10: *# Compute buffer*
11: $\mathbb{B} = \mathbb{V}$
12: **for** $b \in \{0, \dots, B - 1\}$ **do**
13:     $\text{tmp} = \mathbb{B}$
14:     **for** $s \in \text{tmp}$ **do**
15:         $\mathbb{B} = \mathbb{B} \cup \mathbb{N}_s$
16:     **end for**
17: **end for**
18: *# Exclude buffer for training set*
19: $\mathbb{T} = \mathbb{S} \setminus \mathbb{B}$
20: **return** $\mathbb{T}, \mathbb{V}$

---

where $\mathbb{A}$ is the adjacency matrix of the spatial graph; $\mathbb{D}$ is a diagonal matrix with the number of neighbors of each location; $\hat{\rho}$ determines the correlation between an observation and its neighbors, learned from the true residuals (the errors of the predictive model in Step 1); and $\hat{\lambda}$ is chosen to match the variance of the true residuals exactly. We sample $\boldsymbol{R}$ independently to ensure that the synthetic residuals are independent of the treatment, and therefore, by the backdoor criterion (Pearl, 2009), the only possible confounders are those in $\boldsymbol{X}$. The parameter $\hat{\rho}$ by first computing the vector of the neighbors' mean for each node in the graph and then taking the correlation between the nodes and the neighbor's means. Fig. 6 contains a visual example in a spatial map of true and generated residuals and a comparison of the smoothness of the true and synthetic residuals across all `SpaceEnvs`, entailing an almost exact match.

**Code and API**    The codebase for generating an environment is provided separately from the main package at `https://anonymous.4open.science/r/space-data-0C73`. Users can use this codebase to generate new `SpaceEnvs` from existing or new data collections using the steps outlined below. See the repository's documentation for details.

Each environment is specified by a config file using the Hydra framework (Yadan, 2019) and located in the code repository under `conf/spaceenv/` as a `.yaml` file. The generation process starts with a simple command:

```
python train_spaceenv.py spaceenv=<config_name>
```

For example, `config_name=healthd_dmgrcs_mortality_disc` corresponds to the config file illustrated in Fig. 7.

It is possible to use the config file system to generate transforms of variables (for example, to binarize or take logarithms), change the AutoML engine defaults (see Table 5), etc. The training script will try to download the contents of `data_path` (which points to a data collection) and `graph_path` from our Harvard Dataverse repository when not found locally. It also allows users to use their own Datavererse repositories. For instance, larger datasets may require extending the `time_limit` parameter in Table 5. See the repository documentation for details.

```
data_collection: air_pollution_mortality_us
treatment: qd_mean_pm25
outcome: cdc_mortality_pct
covariate_groups:
    - cs_poverty
    - race:
        - cs_hispanic
        - cs_white
    - [...]
```

Figure 7: Config file example: `conf/spaceenv/healthd_dmgrcs_mortality_disc.yaml`.

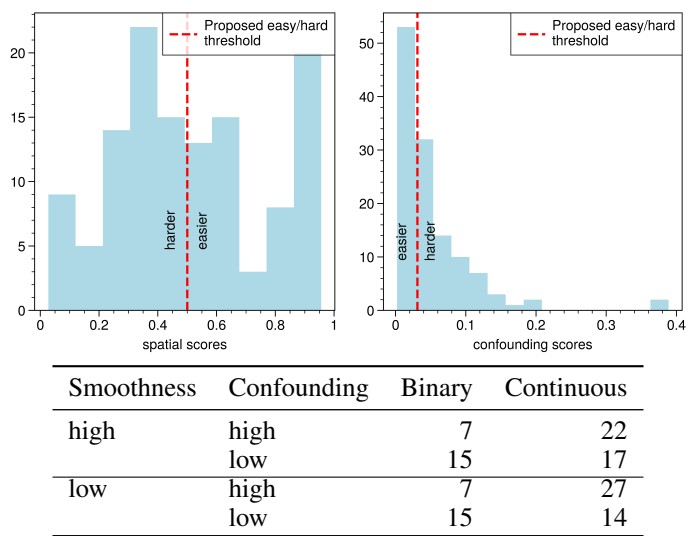

| Smoothness | Confounding | Binary | Continuous |
|---|---|---|---|
| high | high | 7 | 22 |
|  | low | 15 | 17 |
| low | high | 7 | 27 |
|  | low | 15 | 14 |

Figure 8: *(Top left)*: Spatial smoothness scores for all covariates in all environments; *(Top right)*: confounding scores for all covariates in all environments; *(Bottom)*: number of datasets in each combination of low and high smoothness and confounding.

**Sharing and documentation of collections and environments**   The `SpaceEnvs` supported in our Python package are publicly available through our Harvard Dataverse research repository. The metadata for each generated environment is contained in their corresponding entry at our Harvard Dataverse repository, including the full config file for reproducibility with the API explained above. It is also saved as a property of a `SpaceEnv` object when using the Python package.

Each `DataCollection` is documented in the package repository and documentation. We include a description of the data sources, purpose, exploratory data analysis, and other relevant information. The contents of the documentation is inspired by data nutrition labels (Stoyanovich & Howe, 2019). This documentation can be accessed (anonymized) at `https://anonymous.4open.science/r/space-data-0C73`.

**Distribution of spatial and confounding scores**   Fig. 8 shows the distribution of the spatial smoothness and confounding scores (from the exposure-response function) among all covariates of all environments. The median across all datasets is indicated as a vertical red line. The smoothness scores appears to be distributed uniformly across environments, while the confounding scores are heavily skewed.

```
from spacebench import SpaceEnv
env = SpaceEnv('healthd_dmgrcs_mortality_disc')
dataset = env.make()
print(dataset)
```

```
SpaceDataset with a missing spatial confounder:
  treatment: (3109,) (binary)
  confounders: (3109, 30)
  outcome: (3109,)
  counterfactuals: (3109, 2)
  confounding score of missing: 0.02
  spatial smoothness score of missing: 0.11
  graph edge list: (9237, 2)
  graph node coordinates: (3109, 2)
  parent SpaceEnv: healthd_dmgrcs_mortality_disc
WARNING ⚠ : this dataset contains a (realistic) synthetic outcome!
By using it, you agree to understand its limitations.  The variable
names have been masked to emphasize that no  inferences can be made
about the source data.
```

(a) Creating a benchmark environment.

```
from spacebench import DatasetEvaluator
results = your_method(dataset)
metrics = DatasetEvaluator(dataset).eval(
  ate=results['ate'],
  erf=results['erf'],  # exposure response curve
  counterfactuals=results['counterfactuals'],
)
metrics.keys()  # contains error metrics
```

```
dict_keys(['ite_apehe', 'erf_ase', 'ate_se'])
```

(b) Evaluating the performance of a method on a dataset.

Figure 9: Examples of `SpaCE` Python package usage.

## C SpaCE API: ACCESSING ENVIRONMENTS, MAKING BENCHMARK DATASETS, AND EVALUATING THEM

Recall that to obtain a benchmark dataset for spatial confounding, we must 1) create a SpaceEnv which contains real treatment and confounder data, and a realistic semi-synthetic outcome, 2) create a SpaceDataset which masks a spatially-varying confounder and facilitates the data loading pipeline for causal inference. Fig. 9 illustrates the use of the SpaCE package. The top panel illustrates how to generate a SpaceEnv simply by calling the constructor from the environment name (which takes care of downloading the necessary data) and using the make() method to obtain the SpaceDataset (which masks the missing confounder and packages the dataset elements to facilitate data loading for causal inference methods). As with any software tool, the API is subject to change based on feedback from users.

## D HYPER-PARAMETER TUNING AND ADDITIONAL DETAILS OF BENCHMARKS

Implementations of the baselines discussed in Section 5 are given in the SpaCE package under the spacebench.algorithms submodule of the SpaCE python package. They usage is documented in the package documentation. The benchmarks/ folder in the implementation code contains all the scripts to replicate the reported baselines.

For the experiments, we looped over all possible SpaceEnvs and all over possible SpaceDatasets, 124 in total. We implemented automatic hyper-parameters for the relevant baseline models using the library Ray Tune, thereby minimizing human-induced biases. For all but DAPSm, the tuning metric is implemented as the out-of-sample mean-squared error (MSE) from a validation set obtained with the spatially-conscious splitting method of Algorithm 1. After selecting the best hyperparameters, the method was retrained in the full data. One exception is DAPSm, for which the tuning metric was "covariate balance," as originally proposed by the authors. Notice that test MSE is not a metric tailored to causal inference. However, it is reasonable to expect the method that performs less overfitting would also perform better at estimating causal effects in the training data, but this need not always be the case. Unfortunately, no obvious widely applicable hyperparameter selection criteria exist for causal inference. Table 6 summarizes our hyperparameter search space for different baseline models.

| Model | Iterations | Tuning Metric | Value |
|---|---|---|---|
| GCNN | 2,500 | Dimension of the hidden layers (hidden_dim) | 16 or 64 |
| | | Number of hidden layers (hidden_layers) | 1 or 2 |
| | | Weight decay (weight_decay) | 1e-6 to 1e-1 |
| SPATIAL+ | 2,500 | lam_t | loguniform between 1e-5 and 1.0 |
| | | lam_y | loguniform between 1e-5 and 1.0 |
| SPATIAL | 2,500 | lam | loguniform between 1e-5 and 1.0 |
| DAPSM | N/A | propensity_score_penalty_value | Choice among [0.001, 0.01, 0.1, 1.0] |
| | | propensity_score_penalty_type | Choice between l1 and l2 |
| | | spatial_weight | Uniform between 0.0 and 0.1 |
| | | caliper_type | daps |
| | | matching_algorithm | optimal |

Table 6: Hyperparameters tuning for different baseline models. The models are tested with a validation set from spatially aware folds as a breadth-first search with 2% of points and their neighbors, resulting in about 10% of the data.

**Computing and hardware resources** The computations in this paper were run on a Mac OS M1 with ten cores in approximately 24 hours. most of the computation driven by training the graph neural network benchmarks.

## E LICENSES OF NEW AND REUSED ASSETS

We provide the following free and open source resources:

| Resource name | License |
|---|---|
| Python package `SpaCE` [2] | MIT |
| Model-training software repository `space-data` [3] | MIT |
| Benchmark data collection hosted at Harvard Dataverse | CC0 1.0 |

Table 7: Provided new resources

The `DataCollections` used for generating `SpaceEnvs` aggregate various data sources. The licenses of these data sources are summarized in Table 8, which allow sharing and reuse for non-commercial purposes.

| Data source | Reference | License |
|---|---|---|
| Synthetic Medicare Data for Environmental Health Studies | `https://doi.org/10.7910/DVN/L7YF2G` | CC0 1.0 |
| NOAA | `https://www.nauticalcharts.noaa.gov/data/data-licensing.html` | CC0 1.0 |
| CDC | `https://wonder.cdc.gov/DataUse.html` | Public domain |
| U.S. Census Bureau | `https://ask.census.gov/prweb/PRServletCustom?pyActivity=pyMobileSnapStart&ArticleID=KCP-4928` | Public domain |
| The Bureau of Labor Statistics (BLS) | `https://www.bls.gov/opub/copyright-information.html` | Public domain |
| PRISM | `https://prism.oregonstate.edu/terms/` | Open for non-commercial purposes |
| Police shooting by the Washington Post | `https://github.com/washingtonpost/data-police-shootings` | CC BY-NC-SA 4.0 |
| US Broadband Data | `https://github.com/microsoft/USBroadbandUsagePercentages` | Open Use of Data Agreement v1.0 |
| Total PM2.5 | `https://sedac.ciesin.columbia.edu/data/set/aqdh-pm2-5-concentrations-contiguous-us-1-km-2000-2016` | Public domain |
| PM2.5 Components | `https://www.ciesin.columbia.edu/data/aqdh/pm25component-EC-NH4-NO3-OC-SO4-2000-2019/` | Public domain |

Table 8: Major existing reused data resources

## F    ADDITIONAL EXPERIMENT ANALYSIS

This section contains supporting figures and tables for the experiment results presented in Section 5.

- Fig. 10 presents visual outcomes (means and error bars) for various environments. Error bars represent 95% asymptotic normal confidence intervals.

- Table 9 contains the results of a mixed-effects model (Gałecki et al., 2013) to evaluate the interaction between the methods and the confounding and smoothness scores. The dependent variable is the error in causal effect estimation. Specifically, it is the RMISE when estimating the ERF. The independent variables are indicator variables of the spatial confounding methods and their interactions with the smoothness and confounding scores. A positive coefficient means more error, and a negative coefficient means less error. The model controls for random effects by `SpaceDataset`. The $t$-value statistic indicates statistical significance.

See Section 5 for discussion of the results.

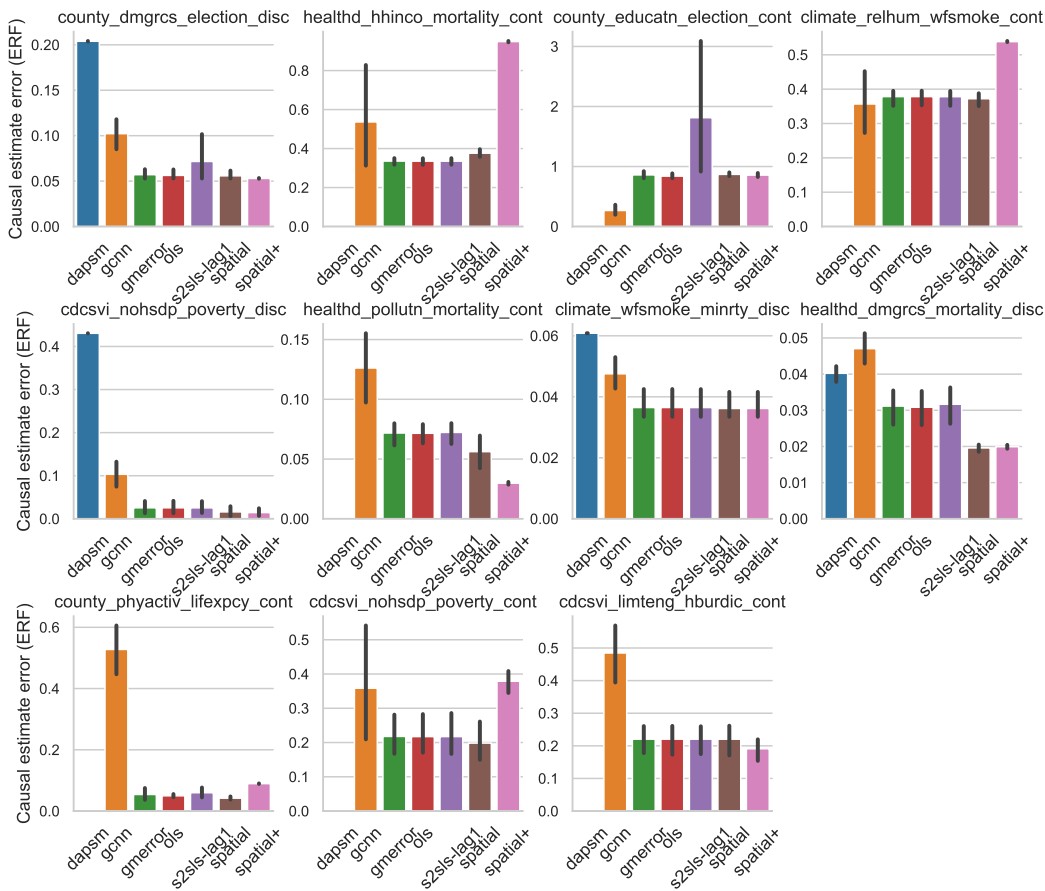

Figure 10: Baseline model results by `SpaceEnv`. Models with lower bars (errors) perform best. The model can be described as

| TERM | COEFFICIENT | STD. ERROR | T-VALUE |
|---|---|---|---|
| (INTERCEPT) | 0.15 | 0.07 | 2.22 |
| SPATIAL | 0.02 | 0.08 | 0.26 |
| SPATIAL+ | 0.00 | 0.08 | 0.06 |
| GCNN | 0.10 | 0.09 | 1.17 |
| DAPSM | 0.78 | 0.19 | 4.13 |
| GMERROR | 0.00 | 0.08 | 0.01 |
| S2SLS | -0.01 | 0.08 | -0.08 |
| OLS:SMOOTHNESS | -0.15 | 0.10 | -1.51 |
| SPATIAL:SMOOTHNESS | -0.19 | 0.10 | -1.88 |
| SPATIAL+:SMOOTHNESS | -0.09 | 0.10 | -0.94 |
| GCNN:SMOOTHNESS | -0.21 | 0.13 | -1.66 |
| DAPSM:SMOOTHNESS | -0.80 | 0.15 | -5.29 |
| GMERROR:SMOOTHNESS | -0.15 | 0.10 | -1.53 |
| S2SLS:SMOOTHNESS | -0.14 | 0.10 | -1.46 |
| OLS:CONFOUNDING | 0.61 | 0.34 | 1.77 |
| SPATIAL:CONFOUNDING | 0.98 | 0.34 | 2.86 |
| SPATIAL+:CONFOUNDING | 1.09 | 0.34 | 3.17 |
| GCNN:CONFOUNDING | 0.73 | 0.37 | 1.96 |
| DAPSM:CONFOUNDING | -2.94 | 2.38 | -1.23 |
| GMERROR:CONFOUNDING | 0.61 | 0.34 | 1.78 |
| S2SLS:CONFOUNDING | 0.63 | 0.34 | 1.83 |

Table 9: Coefficients of a statistical linear model for the error in ERF estimation with random effects per dataset and environment and fixed effects by algorithm and environment complexity (smoothness and confounding).

# G  Q&A

*Q: Why use AutoML for synthetic data generation instead of causal alternatives such as Meta-learners?* Generating a valid `SpaceEnv` does not require that the first step of the AutoML is causal. On the one hand, we need not claim that the counterfactuals of a `SpaceEnv` are causal in the sense of the original data collection. On the other hand, using existing causal tools (like meta-learners) was challenged by the availability of implementations for continuous treatments and individualized effects that do not rely on specific models. AutoML offered several advantages for automatic calibration, ensembling from a large model, etc. Putting everything in the balance, relying on AutoML without Meta-learners seemed to achieve better the goals of reducing human intervention in the data generation and avoiding favoring specific models other than others by using large ensembles.

*Q: Why inferences about the original data sources are not possible if the data is realistic?* There are three main reasons that result from the fundamental problem of causal inference of not observing counterfactuals in real data. The first one is because we do not know if the original data collection includes all relevant confounders; second, as highlighted in the previous answer, the synthetic data generation step need not be causal; third, even if we were using causal methods in the synthetic data generation step, estimation errors and model limitations would mean that we could still not consider the results as a reference for the true causal effect, but simply an estimate. For these reasons, we prefer to strongly advice against such interpretation.

*Q: What measures are you using to avoid data misuse?* Warnings are issued when loading the package and loading environments and datasets in the Python package, as shown in Fig. 9. We also include warning statements in the package documentation and Github repositories.

*Q: Why not learning the joint distribution of* $(\boldsymbol{X}, \boldsymbol{A}, \boldsymbol{Y})$*?*

Sampling from the joint distribution is attractive because it could have the advantage of generating an infinite stream of benchmark datasets from a single data source. However, there could be some limitations with respect to the specific aims of this work. First, we seek to maximize the use of real data because we believe that the complexity of real data is hard to capture in a model. Sampling from the full joint distribution approach would mean using only simulated data, even if grounded on real data. Second, the single 1-dimensional conditional distribution of $(\boldsymbol{Y}|\boldsymbol{X}, \boldsymbol{A})$ is simpler to model than a complex multivariate distribution. Some flexible models allow us to learn complex multivariate distributions, for example. However, relying on a single or a few generative models could artificially favor some causal inference methods in benchmark tasks. Instead, we use an ensemble of diverse predictive models for the 1-dimensional conditional distribution. Last, notice that standard implementations of flexible generative models need not preserve the spatial correlation structure and isolate unobserved confounding (Neal et al., 2020). Therefore, while a joint modeling approach is promising and warrants future investigation, we feel that using the real observed data for the covariates and treatment aligns better with this project's scope.

