# OpenReview forum: "SpaCE: The Spatial Confounding Environment"
_ICLR.cc/2024/Conference — ICLR 2024 poster_

### Official Review · Reviewer_oEAN · 2023-11-02

**Soundness:** 3 good
**Presentation:** 4 excellent
**Contribution:** 3 good
**Rating:** 8
**Confidence:** 3

**Summary:**

The authors focus on the problem of latent spatial confounding, where a number of latent variables that vary smoothly across space causally affect both treatment and outcome.  Some algorithms have been proposed to account for this.  However, evaluation of these algorithms is hindered by a lack of empirical datasets with known ground truth.  To fill this void, the authors propose a method for generating semi-synthetic data with realistic dynamics and spatial confounding.  The spatial variables can then be removed from the training data to create a dataset with latent spatial confounding.  The authors create sets of empirical data in 6 domains, describe the process of generating the semi-synthetic data, and evaluate the performance of multiple spatially-aware causal discovery algorithms across various data settings.  The entire processing framework is encapsulated in a Python package.

**Strengths:**

This is a well-written paper!  As someone familiar with causal modeling but largely unaware of advances in spatial modeling, this paper does a great job of motivating the problem.  While the contribution (generating semi-synthetic data from realistic treatment, outcome and confounders) isn't novel in and of itself, the combination of the extension to a spatial setting, the curation of six data collections, and the creation of a Python package to support the full pipeline are important enough contributions on their own.  I think this paper has a logical flow and explains each piece well, making it a smooth read.  While I can not speak to the algorithms compared, the hyper-parameters varied (smoothness, level of confounding, and binary vs continuous treatment) provide a nice coverage.  I also appreciate the authors' treatment of future work and ethical considerations.

**Weaknesses:**

In Section 2, Figure 2 is described as illustrating that "The closer locations s and s' are, the more correlated X_sMiss and X_s'Miss become.  While Figure 2 does show correlation between two locations s and s', the graphic certainly doesn't show that the correlation increases as s and s' get closer...

Given how information-dense Figure 4 is, I wish more analysis were provided of the results.  As it stands, there is only a single paragraph in Section 5 that discusses findings in Table 4.  In addition, With table 4 being aggregated across all 6 datasets, we're unable to assess how the performance differs across them.  While Figure 10 in the Appendix does show the breakdown by dataset, there is no discussion of the results.  I understand that space is tight, but I think including a bit more analysis in the paper, or at the very least in the appendix, would help the experimental results feel more impactful.

At the end of the second paragraph of Section 2, it says "When a confounder is not unobserved, identification is not guaranteed" - did you mean to say 'When a confounder is not observed" or "unobserved"?

Minor error: in the second to last paragraph of the Introduction, there's the line "SpaCE offers Each dataset has a set of known missing confounders [...]"

**Questions:**

No questions

---

> ### Author Response · Authors · 2023-11-17
>
> We are extremely thankful for your positive feedback. We have made revisions following your suggestions, marked with magenta color in the revised text. We address your comments in order.
>
> 1) We experimented with variations of Fig. 2 but found it the most convenient solution to clarify in the caption that the correlations are stronger as s and s’ get closer.
>
> 2) It is true that we had space limitations and had preferred to expand on clarity at the expense of the analysis of the results. Nonetheless, based on your suggestion, we moved some of the description of data collections in Section 3 to the appendix to make more space for the analysis of the experiments. We thus added two new analysis paragraphs in Section 5.
> The first of these paragraphs presents the results by environment in Fig. 10 of the appendix, providing some new insights, for instance, about the case when the graph neural network outperforms other models. The second paragraph presents a new analysis of the performance by confounding and smoothness score. We use mixed effects regression model adjusting by random effects by environment, allowing us to conclude that higher errors are associated with lower smoothness scores and higher confounding scores as desired. We hope that these new paragraphs increase the impact of the results, as suggested by the reviewer.
>
> 3) Thank you for pointing out the two minor typos. These have been fixed in the revision.

---

> > ### Author Response · Authors · 2023-11-22
> >
> > Dear reviewer, as the author/reviewer discussion period ends today, we would be very grateful if you could indicate if your comments, particularly regarding the additional analysis, have been addressed. We are understanding that the deadline is very tight.

---

### Official Review · Reviewer_189J · 2023-11-03

**Soundness:** 2 fair
**Presentation:** 3 good
**Contribution:** 3 good
**Rating:** 6
**Confidence:** 3

**Summary:**

The authors introduce SpaCE, to provide benchmark datasets and tools for systematically evaluating causal inference methods designed to alleviate spatial confounding. In SpaCE, each dataset includes training data, true counterfactuals, a spatial graph with coordinates, and smoothness and confounding scores. It also includes realistic semi-synthetic outcomes and counterfactuals, generated using state-of-the-art machine learning ensembles, following best practices for causal inference benchmarks.

**Strengths:**

The paper targets a meaningful problem and tries to provide a solution. The structure of the paper is good, and code is provided. The authors provide examples and experiment results to show the effectiveness of the dataset.

**Weaknesses:**

The reviewer has several concerns that need the authors to address.

1. In equation 1. if R_s is the autocorrelation with neighbors, why do the authors consider only an additive model? What if Y = f(X, a, R)? Does the method fail in the cases of Y = f(X, a, R)?

2. Counterfactual inference/generation typically starts from the abduction step that infers the value or distribution of exogenous variables. However, in causal model (1), it is not clear which letter(s) denotes the exogenous variable(s).

3. What is the relationship between the residual R and the exogenous variable in the generation of Y? If R is the exogenous variable for Y, the authors need to consider a more general causal mechanism, rather than just the additive noise model (ANM). If R is not the exogenous variable, why the authors can use equation 1 for counterfactual inference?

**Questions:**

See the weaknesses.
If the authors address my concerns, I will increase the score.

---

> ### Author Response · Authors · 2023-11-16
>
> We are thankful for your positive remarks on the structure of the paper and your accurate summary. We have made improvements in Section 4 of the revision based on your comments, marked with red color.
>
> Perhaps there is still some confusion about the counterfactual generation procedure. The process can be summarized in four steps: First, we learn $f$ that best predicts $Y_s$ from $(X_s, A_s)$ using AutoML. Second, we estimate the empirical additive errors $\hat{R}\_s = Y_s - f(X_s, A_s)$ and their joint (spatial) distribution $(\hat{R}\_s)\_{s\in\mathbb{S}} \sim P_R$. Third, we replace these endogenous empirical residuals with an independent similarly distributed exogenous noise $(R\_s)\_{s\in\mathbb{S}} \sim P_R$. Finally, we obtain counterfactuals by varying the treatment while holding constant the confounders and the exogenous noise. We added this summary in Section 4 for additional clarity.
>
> We answer your questions directly.  As quick remarks,  $R_s$ is indeed the exogenous noise (Q3), and the abduction step is indeed what the reviewer suggests as the typical thing to do (Q2).
>
> The reviewer is concerned about the limitations of additive noise models (ANMs) (Q1, Q3). First, notice that while eq. (1) is additive, SpaCE's benchmark datasets can exhibit complex forms of non-additive noise due to the interactions with the missing confounder, which becomes part of the noise when masked.  The post-masking causal model can be written as $\tilde{Y}_s=f(X^\text{obs}_s, A_s, U_s)$ where the effect of the noise $U_s=(R_s, X_s^\text{miss})$ is likely not additive due to interactions of $X_s^\text{miss}$. Second, it is worth noticing that ANMs remain the focus and leading open challenge in the spatial confounding literature that our paper builds upon (c.f., [2,3,4,5]).  Are the reviewer's concerns motivated by the use and limitations of ANMs in causal discovery tasks [1, 6]? The spatial confounding literature we build upon mainly concerns causal effect estimation, not discovery. While there can be challenges for cause-effect estimation under non-additive noise models too, simple techniques complementing spatial confounding methods for ANMs have been used; for example,  with generalized linear models (e.g., Poisson regression) or outcome transformations (e.g., logarithms) for count-based or binary outcomes [5]. Thus, SpaCE's benchmark datasets are highly relevant to the current state of the art in spatial confounding adjustment.
>
> Regarding potential failures (Q1), the ANM will be a sensible model when the empirical residuals are approximately symmetric and continuously supported, as exemplified in Fig. 3c. We have documented the outcome transformations and stored histograms of the empirical and synthetic residuals with each environment in the Harvard Dataverse platform to ensure validity. We have also emphasized this in the revision.
>
> *References*
>
> - [1] Janzing & Steudel, 2010. “Justifying additive noise model-based causal discovery via algorithmic information theory”
> - [2] Kamal et al. 2023. “Spatial Causality: A Systematic Review on Spatial Causal Inference”. *Geographic Analysis*.
> - [3] Dupont et al. 2022. “Spatial+: a novel approach to spatial confounding”
> - [4] Paciorek, 2010. “The importance of scale for spatial-confounding bias and precision of spatial regression estimators”
> - [5] Urdangarin et al. 2023. “Evaluating recent methods to overcome spatial confounding”
> - [6] Peters et al. 2004. "Causal Discovery with Continuous Additive Noise Models."

---

> > ### Author Response · Authors · 2023-11-21
> >
> > Dear reviewer, thanks again for your review. As the author/discussion period ends tomorrow, we would be extremely grateful if you could let us know if your concerns have been resolved and of other follow-up questions you may have.

---

> > ### Comment · Reviewer_189J · 2023-11-22
> > **Response to reviewer's comment**
> >
> > The authors' comments have addressed my concerns, and I would like to increase the score to 6.

---

### Official Review · Reviewer_ykyy · 2023-11-05

**Soundness:** 3 good
**Presentation:** 4 excellent
**Contribution:** 2 fair
**Rating:** 6
**Confidence:** 4

**Summary:**

The authors describe a framework for evaluating methods for causal inference in spatial data.

**Strengths:**

The authors propose an environment for producing realistic benchmark data sets. Opinions may differ about whether this is a significant contribution, but it is certainly an important step toward verifiable research progress in causal inference.

Despite the problems outlined below, this paper is still worth accepting. The paper itself is well-written and detailed, and the work itself provides valuable infrastructure for later research.

**Weaknesses:**

The decision to defer work on interference is dubious. There are many ways in which the treatments, outcomes, and covariates of neighboring units can interact, including treatment of one unit causing outcome in neighboring units, treatments causing treatments, and outcomes causing outcomes (and even outcomes causing treatments). It would seem reasonable to construct semi-synthetic data generation methods that could, in principle, produce all of these effects.

The authors produce a variety of data sets that use real treatments and covariates (and synthetic outcomes). Given that real treatments are used, there clearly could be spatial correlations among those variables. Thus, generated data sets may already have some degree of treatment-to-treatment spillover or covariate to (multiple) treatment dependence that produces that spatial dependence among treatment values.

The authors do not cite a large and recent literature on evaluation methods for causal inference. While this literature is not about spatial confounding specifically, it provides a large number of analogs in the context of non-spatial confounding and it is worth citing. These include:

Dorie, V., Hill, J., Shalit, U., Scott, M., & Cervone, D. (2019). Automated versus Do-It-Yourself Methods for Causal Inference. Statistical Science, 34(1), 43-68.

Gentzel, A. M., Pruthi, P., & Jensen, D. (2021, July). How and why to use experimental data to evaluate methods for observational causal inference. In International Conference on Machine Learning (pp. 3660-3671). PMLR.

Cheng, L., Guo, R., Moraffah, R., Sheth, P., Candan, K. S., & Liu, H. (2022). Evaluation methods and measures for causal learning algorithms. IEEE Transactions on Artificial Intelligence, 3(6), 924-943.

**Questions:**

Why not include various types of effects (see above) besides confounding?

---

> ### Author Response · Authors · 2023-11-16
>
> Thank you for your positive notes on the paper's strengths. We now address your other comments and questions. The text in the revision addressing these comments will be highlighted in blue.
>
> We included your recommended citations for works related to causal evaluation (outside spatial confounding) in the introduction.
>
> Regarding the deferral of interference to future work, this decision came after careful consideration. The three main reasons are the lack of ensembling methods allowing for spill-overs, the need for spill-over-specific metrics, and the different dataset domains typically considered in the literature. We clarify below:
>
> *Lack of ensembling and ML techniques for interference*. The literature has yet to significantly develop ML models for counterfactual generation under interference. Most existing approaches typically rely on limited manually constructed spill-over exposure maps characterizing interference (e.g., assuming that the nature of interference depends only on the average treatment of neighbors) [1,4]. Graph neural networks (GNNs) are promising to reduce the dependency on manually constructed features [2], but extensive evaluation and research justifying their correct utilization is still needed [3]. Even if considering GNNs, a core design element in SpaCE uses AutoML (including boosting, random forests, NNs, etc.) to avoid artificially favoring a specific machine learning model. Currently, generated synthetic datasets do not exhibit spill-overs since, conditional on the confounders, AutoML baselines do not allow local outcomes to depend on non-local treatments.
>
> *Contrasting estimands and metrics*. The focus of interference research is on estimating spill-over/indirect effects vs direct effects [1]. Spill-over effects, as typically defined [4], are specified in terms of assumptions in changes in the distribution of the treatment of neighbors. These estimands and associated metrics must be defined and explained in detail, deviating from those used for confounding. Tests and evaluations for interference would also require different algorithmic baselines. Addressing these differences within the space limitations of a single conference paper would have required a significant deviation at the expense of clarity and focus on the increasingly important topic of spatial confounding.
>
> *Different focus on the nature of the data*. It is true that, as pointed out by the reviewer, some of our raw data collections could exhibit spillovers. However, the interference literature has more frequently focused on other types of networked data, such as social networks [1, 5], which have traditionally been outside the main concern of the spatial confounding literature. Based on the reference recommended by the reviewer, Cheng et al. (2022), it can also be pointed out that some datasets for interference already exist based on social networks, while no current solution using realistic representative data exists for spatial confounding, which is the important gap filled by SpaCE.
>
> We have included a summarized version of this discussion in section 6.
>
> *References*
>
> - [1] Forastiere et al., 2021. “Forastiere, Laura, Edoardo M. Airoldi, and Fabrizia Mealli. "Identification and estimation of treatment and interference effects in observational studies on networks."
> - [2] Ma & Volker., 2021. “Causal inference under networked interference and intervention policy enhancement”
> - [3] Jian & Sun., 2022. “Estimating Causal Effects on Networked Observational Data via Representation Learning”.
> - [4] Hudgens & Halloran, 2008. “Toward causal inference with interference.”
> - [5] Cheng et al. 2022. “Evaluation methods and measures for causal learning algorithms.”

---

> ### Author Response · Authors · 2023-11-22
>
> Dear reviewer, as the author/reviewer discussion period ends today, we would be very grateful if you could indicate if you have additional questions. We are understanding that the deadline is very tight. We also highlight to the reviewer the additional analysis and experiments indicated in color text in the revision and summarized in the comment to all reviewers [here](https://openreview.net/forum?id=D9rJdtmIG6&noteId=oLoyfbjF9G).

---

### Author Response · Authors · 2023-11-12
**To all reviewers. Initial response.**

We thank you for your thoughtful comments, questions, and suggestions. The reviewers agreed that the paper takes an important step towards verifiable research in causal inference (R1), addresses a meaningful problem in causal inference (R2), and fills a gap in the evaluation of spatial confounding algorithms (R3). The reviewers commented positively on the clarity and structural and logical flow of the paper (R2, R3), the provision of dataset, code, and software tools (R2, R3), and the experimental setup (R3).

We summarize some of the main concerns. R1 points out missing citations that, while outside the spatial confounding literature, need citation since they pertain to the evaluation of casual methods. R1 also wondered about postponing interference/spill-over problems to future work (R1). R2 raised a question about using an additive noise model in the data-generating mechanism and suggested clarifications about the exogenous components of the causal model (1). R3 identified some minor typographical errors (we apologize sincerely), suggested an improvement to Fig. 4, and recommended expanding the analysis of the experimental results in the appendix to increase their impact.

We will begin addressing your concerns in the revision, post a summary of the changes to all reviewers, and provide detailed individual responses to your comments.

---

> ### Author Response · Authors · 2023-11-21
> **Response summary to all reviewers**
>
> Thank you again for your thoughtful comments and suggestions that improved our work during the revision. Answers to all questions raised by the reviewers were posted a few days ago. The revised text indicates the changes with colored lettering.
>
> If the reviewers have remaining questions, we would be very grateful if they could indicate so before the author/reviewer discussion period ends tomorrow.
>
> For your convenience, we summarize our main revisions here. Please refer to the individual responses for additional explanation and other comments not mentioned here.
> * Citations about evaluation protocols in causal inference outside the spatial confounding literature have been added to the introduction and metrics section (R1).
> * A more complete discussion in Section 6 of the additional orthogonal challenges required for spill-over/interference analysis was provided to justify the deferral to future work (R1).
> * We have provided additional justification in Section 4 for using a generative non-linear additive noise model and explaining why with this choice SpaCE's benchmark datasets represent the current state of the art and concerns in the spatial confounding adjustment literature we build upon (R2).
> * Expanded and improved the explanation of the data-generating model in Section 4, particularly the role of the exogenous noise $R_s$ (R2).
> * Significantly expanded the analysis of experiment results in Section 5, adding two new paragraphs with an analysis of the performance by environment and confounding/spatial scores (R3).
> * Fixed minor typos and improved the caption of Fig. 2 (R3).

---

### Meta-Review · Area_Chair_47Rd · 2023-12-08

**Metareview:**

This paper provides a benchmark for causal inference in situations where spatial confounding is present. While its assumptions are somewhat questionable, as noted by the reviewers, such an effort is quite important to share in causal inference research, where the absence of benchmarks hinders research progress more than in other fields.

**Justification For Why Not Higher Score:**

The content is worth sharing, but the topic is not very broad.

**Justification For Why Not Lower Score:**

The existence of benchmarks is useful to encourage the development of causal inference research.

---

### Decision · Program_Chairs · 2024-01-16

Accept (poster)